# From Coarse to Fine: Deep Prototype Refinement Network for Few-Shot Point Cloud Semantic Segmentation

**Changshuo Wang** [1]  **Weijun Li** [2]  **Shuting He** [3]  **Xiang Fang** [4]  **Xingyu Gao** [5]  **Zhonghang Liu** [6]  **Prayag Tiwari** [7]
**Dimitrios Kanoulas** [1]

## Abstract

Few-shot point cloud semantic segmentation (**FS-PCSS**) aims to achieve precise segmentation of novel categories using only limited labeled samples. Existing prototype-based methods rely on shallow feature fusion, failing to adequately model the feature distribution shift between support and query sets, resulting in insufficient prototype adaptation. To address this, we propose the **D**eep **P**rototype **R**efinement **Net**work (**DPR-Net**), which systematically achieves progressive adaptation by constructing a coarse-to-fine prototype evolution trajectory. Our core **D**ynamic **P**rototype **R**efinement (**DPR**) module decomposes features into common and distinctive subspaces based on channel activation, enabling targeted adjustment of domain-sensitive features while preserving class-shared semantics. By cascading multiple refinement modules, we construct a prototype trajectory transitioning from support-biased to query-adapted representations, mitigating both under- and over-adaptation. Furthermore, our **M**ixture **o**f **P**rototype **E**xperts (**MoPE**) mechanism treats prototypes at different stages as experts and ensembles their predictions through confidence-driven weighting. Extensive experiments demonstrate state-of-the-art performance with high efficiency. Our code will be available at https://github.com/changshuowang/DPR-Net.

[1]Department of Computer Science, University College London, London, United Kingdom [2]Institute of Semiconductors, Chinese Academy of Sciences, Beijing, China [3]School of Computing and Artificial Intelligence, Shanghai University of Finance and Economics, Shanghai, China [4]ERI@N, Interdisciplinary Graduate Programme, Nanyang Technological University, Singapore [5]Institute of Microelectronics, Chinese Academy of Sciences, Beijing, China [6]SmartMore Corporation Ltd., Shenzhen, China [7]School of Information Technology, Halmstad University, Sweden. Correspondence to: Weijun Li <wjli@semi.ac.cn>.

*Proceedings of the 43rd International Conference on Machine Learning*, Seoul, South Korea. PMLR 306, 2026. Copyright 2026 by the author(s).

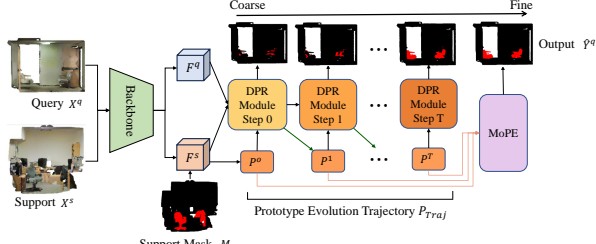

*Figure 1.* **Overview of DPR-Net.** Given query point cloud $X^q$ and support set $X^s$ with mask $M$, we extract features $F^q$ and $F^s$ through a shared backbone. The support mask is applied via masked average pooling to obtain the initial coarse prototype $P^0$. Our model constructs a coarse-to-fine evolution trajectory $P_{Traj} = \{P^0, P^1, \ldots, P^T\}$ through $T$ cascaded DPR modules. Finally, the MoPE module adaptively ensembles all intermediate prototypes through confidence-driven weighting, producing the final segmentation output $\hat{Y}^q$.

## 1. Introduction

Point cloud semantic segmentation (Wang et al., 2022) assigns each point to predefined semantic categories, serving as critical technology for autonomous driving (Zhao et al., 2023; Chib & Singh, 2023), robotic navigation (Soori et al., 2023; Goel & Gupta, 2020), and virtual reality (Devagiri et al., 2022; Sereno et al., 2020). While traditional methods achieve remarkable progress with large-scale annotated data, the annotation cost for point cloud data remains prohibitively high, particularly for rare categories, severely constraining deployment in emerging applications.

Few-shot point cloud semantic segmentation (FS-PCSS) (Wang et al., 2026; Li et al., 2024; Xiong et al., 2024) addresses this bottleneck by enabling models to segment novel categories using only limited labeled support samples. Current methods (Zhao et al., 2021b; Mao et al., 2022) predominantly follow the prototype learning paradigm: extracting features via encoders (e.g., DGCNN (Wang et al., 2019)), generating initial prototypes through masked average pooling, and refining them by incorporating query information. However, existing refinement strategies (Lai et al., 2022; Zhu et al., 2023) remain superficial—they typically perform simple feature fusion without adequately modeling the fea-

ture distribution shift between support and query sets. When distribution gaps exist (e.g., variations in scanning density, occlusions, or class appearance), these shallow approaches fail to effectively bridge the shift, limiting adaptation performance and generalization to novel categories.

The fundamental challenge in FS-PCSS lies in achieving effective adaptation from support to query representations through progressive refinement rather than crude one-step fusion. Inspired by the success of iterative refinement paradigms in various domains, we reformulate prototype adaptation as a principled multi-stage refinement process that bridges the distribution shift. As illustrated in Fig. 1, instead of directly jumping from coarse support prototypes to final predictions, our approach constructs a prototype evolution trajectory through cascaded refinement modules, enabling progressive adaptation that balances feature preservation and distribution alignment.

To this end, we propose the **D**eep **P**rototype **R**efinement **Net**work (DPR-Net), which systematically achieves progressive adaptation by constructing a coarse-to-fine prototype evolution trajectory. Our core **D**ynamic **P**rototype **R**efinement (DPR) module decomposes features into common and distinctive subspaces based on channel activation, explicitly disentangling shared semantic features from domain-sensitive variations for targeted adjustment. Through cascaded refinement with multiple intermediate steps, we construct a prototype trajectory $P_{Traj} = \{P^0, P^1, \ldots, P^T\}$ that transitions from initial support-biased representations $P^0$ to query-adapted prototypes $P^T$, avoiding both insufficient adaptation and overfitting. Furthermore, our **M**ixture **o**f **P**rototype **E**xperts (MoPE) mechanism treats prototypes at different refinement stages as complementary "experts", leveraging trajectory diversity through confidence-driven weighting for robust predictions. Each intermediate prototype captures a unique balance between support priors and query adaptation, enabling adaptive point-wise ensemble.

Our main contributions are as follows:

- We propose DPR-Net, which reformulates prototype adaptation in FS-PCSS as a principled progressive refinement process, providing a systematic coarse-to-fine paradigm that explicitly models the support-query distribution shift through cascaded refinement modules.

- We present the DPR module with dual-subspace decomposition, which disentangles shared semantics from domain-sensitive variations and enables targeted adaptive adjustment at each refinement stage.

- We design the MoPE mechanism for confidence-driven multi-stage prototype ensembling, fully leveraging complementary information across the refinement trajectory for robust segmentation.

- Extensive experiments on S3DIS and ScanNet demonstrate state-of-the-art performance. With only 0.28M parameters, DPR-Net achieves 80.76% mIoU on S3DIS (2-way 1-shot).

## 2. Related Work

### 2.1. Point Cloud Semantic Segmentation

Point cloud semantic segmentation (Wang et al., 2022; Wu et al., 2022), as a core task of 3D scene understanding, aims to assign semantic labels to every point. Early methods, represented by PointNet (Qi et al., 2017a), directly processed point clouds using shared MLPs and global pooling but were limited in local feature extraction. Subsequently, PointNet++ (Qi et al., 2017b) introduced hierarchical sampling to enable layer-wise learning, while DGCNN (Wang et al., 2019) proposed the EdgeConv operation to dynamically construct local graphs and capture inter-point relationships, becoming a classic backbone. Recent research has progressed along three main directions: (1) Transformer architectures (e.g., Point Transformer (Zhao et al., 2021a)) model long-range dependencies; (2) Mamba-based models (e.g., PointMamba (Liang et al., 2024)) achieve efficient encoding through State Space Models; (3) self-supervised pre-training methods (e.g., PointGPT (Chen et al., 2023), PointMAE (Pang et al., 2023)) leverage masked modeling to reduce annotation requirements. Additionally, RandLA-Net (Hu et al., 2020) optimizes large-scale processing, and multi-modal fusion approaches (e.g., PointCLIP (Zhang et al., 2022)) enhance robustness. Despite their excellent performance, fully-supervised methods rely heavily on large-scale annotated data, making them difficult to transfer to few-shot scenarios.

### 2.2. Few-Shot Point Cloud Semantic Segmentation

Few-Shot Point Cloud Semantic Segmentation (FS-PCSS) (Mao et al., 2022; Zhang et al., 2023; He et al., 2023; An et al., 2025) addresses data scarcity by generating prototypes from limited support samples to segment novel categories in query point clouds. Built upon meta-learning, early work AttMPTI (Zhao et al., 2021b) established the foundation with pre-trained DGCNN (Wang et al., 2019), multi-prototype generation, and label propagation. Subsequent advances focus on three main directions: *feature enhancement*, *prototype optimization*, and *architectural innovation*. For feature enhancement, methods like BFG (Mao et al., 2022) employ bi-directional globalization to fuse cross-set features. For prototype optimization, DENet (Xiong et al., 2024) uses bi-directional aggregation to mitigate intra-class diversity and purify ambiguous prototypes, while GPCPR (Wei et al., 2025) fuses LLM-generated category descriptions with pseudo query context to alleviate bias. For architectural innovation, DyPolySeg (Wang et al.,

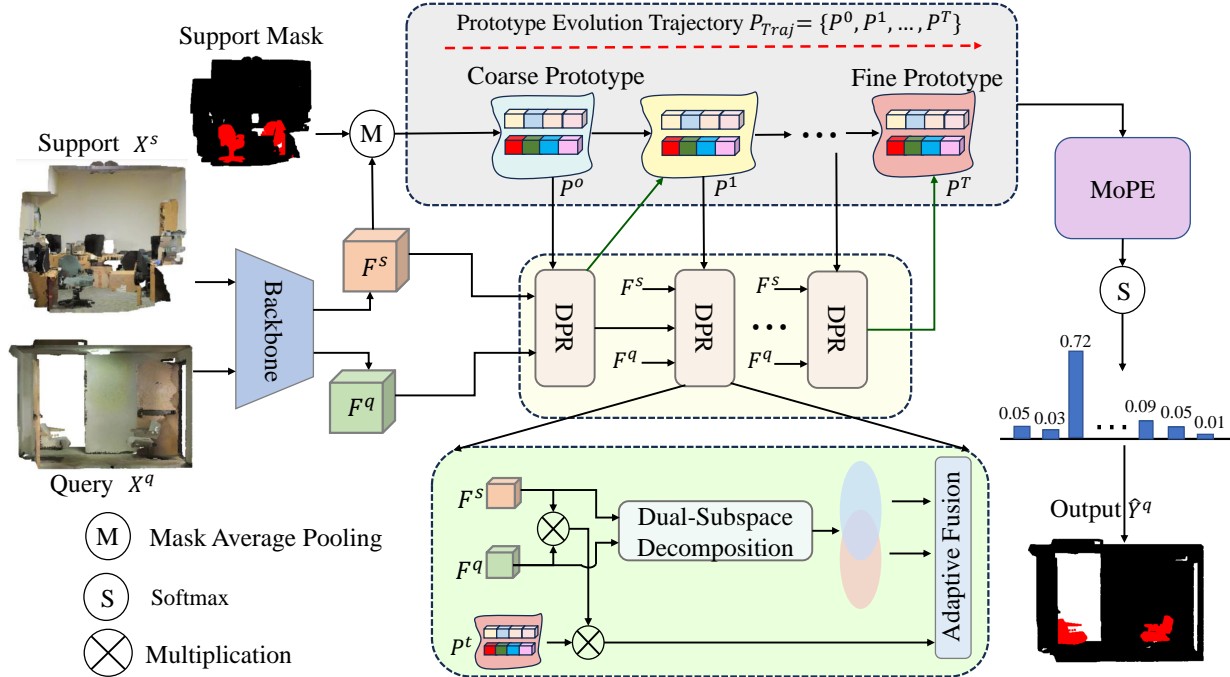

*Figure 2.* **Overall architecture of DPR-Net.** Given a query point cloud $X^q$ and support set $X^s$ with mask $M$, we first extract features $F^q$ and $F^s$ through a shared backbone network. The support mask is applied via masked average pooling to obtain the initial coarse prototype $P^0$. Our model then constructs a refinement trajectory $P_{Traj} = \{P^0, P^1, \ldots, P^T\}$ through $T$ cascaded Dynamic Prototype Refinement (DPR) modules. Each DPR module takes support features $F^s$, query features $F^q$, and the previous prototype $P^{t-1}$ as input, performing dual-subspace decomposition to disentangle shared semantics from domain-sensitive variations, followed by adaptive fusion to generate the refined prototype $P^t$. The bottom panel details the DPR module structure: support and query features are element-wise multiplied with the prototype, then decomposed into common and distinctive subspaces for targeted processing. Finally, the Mixture of Prototype Experts (MoPE) module adaptively ensembles all intermediate prototypes $\{P^0, P^1, \ldots, P^T\}$ through confidence-driven weighting, producing per-point classification probabilities via softmax to generate the final segmentation output $\hat{Y}^q$.

2025a) combines dynamic polynomial convolution with Mamba to capture both local and global features. Additionally, Seg-PN (Zhu et al., 2024) and TaylorSeg (Wang et al., 2025c) forgo pre-training to reduce domain shift, while COSeg (An et al., 2024) enhances practicality by expanding input points to mitigate information leakage. Despite these advances, existing methods rely on superficial feature fusion for prototype refinement, failing to systematically model the distribution relationship between support and query sets.

# 3. Methodology

In this section, we first define the task formulation of FS-PCSS, then introduce the proposed DPR-Net architecture (as shown in Fig. 2). Subsequently, we elaborate on its core component—the Dynamic Prototype Refinement (DPR) module, followed by the progressive prototype refinement process and the Mixture of Prototype Experts (MoPE) mechanism. Finally, we present our training objectives.

## 3.1. Problem Formulation

FS-PCSS adopts the episodic learning paradigm, partitioning the dataset into seen classes $\mathcal{C}_{seen}$ and unseen classes $\mathcal{C}_{unseen}$. Each few-shot task is as a $N$-way $K$-shot problem, comprising a support set $\mathcal{S}$ and a query set $\mathcal{Q}$.

Formally, given a support set $\mathcal{S} = \{(\mathbf{X}_i^s, \mathbf{Y}_i^s)\}_{i=1}^{N \times K}$ containing $N$ classes ($N$-way) with $K$ labeled samples per class ($K$-shot), where each sample $\mathbf{X}_i^s \in \mathbb{R}^{M_s \times 3}$ consists of $M_s$ points with 3D coordinates and $\mathbf{Y}_i^s \in \{0,1\}^{M_s}$ denotes binary masks, and a query set $\mathcal{Q} = \{\mathbf{X}^q, \mathbf{Y}^q\}$ where $\mathbf{X}^q \in \mathbb{R}^{M_q \times 3}$ and $\mathbf{Y}^q \in \{0, 1, \ldots, N\}^{M_q}$, our objective is to predict semantic labels for each point in the query point cloud, classifying them into $N$ target classes and 1 background class (where 0 represents background).

## 3.2. Overall Architecture of DPR-Net

Our proposed DPR-Net consists of four synergistic components (see Fig. 2): (1) a feature encoder $\Phi(\cdot)$ that extracts deep feature representations from input point clouds; (2) an initial prototype extraction module that computes coarse prototypes $\mathbf{P}^0 \in \mathbb{R}^{(N+1) \times D}$ from the support set (including

background class); (3) the Dynamic Prototype Refinement (DPR) module as the core component that progressively refines prototypes through $T$ iterative steps, constructing a refinement trajectory $P_{Traj} = \{\mathbf{P}^0, \mathbf{P}^1, \ldots, \mathbf{P}^T\}$; and (4) the Mixture of Prototype Experts (MoPE) mechanism that adaptively ensembles predictions from all prototypes along the trajectory to generate the final segmentation result.

For the feature encoder, we adopt the non-parametric model Seg-NN (Zhu et al., 2024) as our backbone network to extract deep features $\mathbf{F}^q \in \mathbb{R}^{M_q \times D}$ and $\mathbf{F}^s \in \mathbb{R}^{N \times K \times M_s \times D}$ from the query point cloud $\mathbf{X}^q$ and support point clouds $\mathbf{X}^s$, respectively, where $D$ denotes the feature dimension. The initial prototype $\mathbf{P}^0$ is obtained by averaging the support features $\mathbf{F}^s$, serving as the starting point for the refinement process.

The core refinement process is realized by stacking $T$ DPR modules. At time step $t$ ($t = 0, \ldots, T-1$), the DPR module $\mathcal{D}_\theta$ receives the prototype from the previous step $\mathbf{P}^t$, query features $\mathbf{F}^q$, and support features $\mathbf{F}^s$, and outputs the refined prototype $\mathbf{P}^{t+1}$:

$$\mathbf{P}^{t+1} = \mathcal{D}_\theta(\mathbf{P}^t, \mathbf{F}^q, \mathbf{F}^s), \quad t = 0, 1, \ldots, T-1. \quad (1)$$

This iterative process generates a prototype refinement trajectory $P_{Traj} = \{\mathbf{P}^0, \mathbf{P}^1, \ldots, \mathbf{P}^T\}$, where each prototype $\mathbf{P}^t$ represents a different intermediate stage of adaptation to the query set. Finally, we employ the MoPE mechanism to fuse predictions from all prototypes along the trajectory $P_{Traj}$, generating the final segmentation.

### 3.3. Dynamic Prototype Refinement Module

The DPR module is the core component that enables progressive prototype refinement. **Motivation:** In few-shot learning, not all feature channels contribute equally to generalization. Certain channels encode class-invariant geometric semantics (e.g., object shape structures) that remain stable and should be preserved, while other channels capture domain-sensitive appearance features (e.g., scanning density, noise patterns) that require adaptive adjustment. To address this, we achieve targeted refinement by decomposing features into dual subspaces that disentangle shared semantics from domain-sensitive variations.

**Channel Activation Analysis.** To identify important channels in both support and query sets, we first compute channel-level activation maps. For a feature $\mathbf{F} \in \mathbb{R}^{M \times D}$ (where $M$ is the number of points and $D$ is the feature dimension), its channel activation map $\mathbf{A} \in \mathbb{R}^D$ is computed through global average pooling followed by activation normalization:

$$\mathbf{A} = \sigma(\text{GAP}(\mathbf{F})), \quad (2)$$

where $\sigma(\cdot)$ denotes the sigmoid function and $\text{GAP}(\cdot)$ represents global average pooling over the spatial dimension. We compute the query channel activation map

$\mathbf{a}^q = \sigma(\text{GAP}(\mathbf{F}^q)) \in \mathbb{R}^D$ and prototype activation map $\mathbf{a}^p = \sigma(\mathbf{P}^t) \in \mathbb{R}^{(N+1) \times D}$.

**Dual-Subspace Decomposition.** Based on the channel activation information, we introduce a threshold $\tau \in (0, 1)$ (initialized to 0.5) to determine channel activation, thereby decomposing the $D$-dimensional feature space into three complementary subspaces:

$$\begin{aligned}
\mathcal{C}^t &= \{d \mid \mathbf{a}^q_d > \tau \text{ and } \mathbf{a}^p_{c,d} > \tau, \forall c\}, \\
\mathcal{U}^t_q &= \{d \mid \mathbf{a}^q_d > \tau \text{ and } \mathbf{a}^p_{c,d} \leq \tau\}, \quad (3) \\
\mathcal{U}^t_p &= \{d \mid \mathbf{a}^q_d \leq \tau \text{ and } \mathbf{a}^p_{c,d} > \tau\},
\end{aligned}$$

where $\mathcal{C}^t$ is the common subspace consisting of channels co-activated in both query and prototype, representing shared semantics; $\mathcal{U}^t_q$ is the query-distinctive subspace containing query-specific discriminative information; and $\mathcal{U}^t_p$ is the prototype-distinctive subspace containing prototype-specific discriminative information that must be preserved.

**Targeted Fusion Strategy.** Based on subspace decomposition, we adopt differentiated processing strategies. For the common subspace $\mathcal{C}^t$, we employ a conservative fusion mechanism to preserve shared semantic information:

$$\mathbf{G}^t_\mathcal{C} = \left(\frac{\mathbf{a}^q + \mathbf{a}^p}{2}\right) \odot \mathbb{k}_{\mathcal{C}^t}, \quad (4)$$

where $\odot$ denotes element-wise multiplication and $\mathbb{k}_{\mathcal{C}^t} \in \{0, 1\}^D$ is a binary mask. This ensures that refined prototypes maintain class-discriminative semantic information.

For distinctive subspaces, we adopt an adaptive fusion mechanism:

$$\mathbf{G}^t_\mathcal{U} = \left(\frac{\mathbf{a}^q \odot \mathbb{k}_{\mathcal{U}^t_q} + \mathbf{a}^p \odot \mathbb{k}_{\mathcal{U}^t_p}}{2}\right), \quad (5)$$

where the query's distinctive features $\mathbf{a}^q \odot \mathbb{k}_{\mathcal{U}^t_q}$ provide guidance for prototype evolution, while the prototype's distinctive features $\mathbf{a}^p \odot \mathbb{k}_{\mathcal{U}^t_p}$ preserve its unique representation.

**Cross-Attention Enhancement.** To further enhance the prototype's perception of query spatial context, we introduce a cross-correlation attention mechanism. We first project query and support features into a low-dimensional space: $\mathbf{Q}_{proj} = \text{Conv}_{1\times1}(\mathbf{F}^q) \in \mathbb{R}^{M_q \times d}$ and $\mathbf{S}_{proj} = \text{Conv}_{1\times1}(\mathbf{F}^s) \in \mathbb{R}^{M_s \times d}$, where $d = 72 < D$ to reduce computational complexity. Then we compute scaled dot-product attention:

$$\mathbf{C}_{cross} = \text{softmax}\left(\frac{\mathbf{Q}_{proj} \cdot \mathbf{S}^\top_{proj}}{\sqrt{d}}\right) \in \mathbb{R}^{M_q \times M_s}. \quad (6)$$

This cross-correlation matrix $\mathbf{C}_{cross}$ captures fine-grained similarity relationships between query and support points. We utilize this attention matrix to perform weighted aggregation: $\mathbf{P}^t_{cross} = \mathbf{C}_{cross} \cdot \text{Proj}_\phi(\mathbf{P}^t)$, where $\text{Proj}_\phi$ is a learnable linear projection.

**Prototype Refinement.** We adaptively fuse the three information sources—common features $\mathbf{G}_{\mathcal{C}}^t$, distinctive features $\mathbf{G}_{\mathcal{U}}^t$, and cross-enhancement $\mathbf{P}_{cross}^t$. We introduce a balancing coefficient $\alpha \in (0, 1)$ (set to 0.5) to control the relative importance:

$$\tilde{\mathbf{P}}^{t+1} = \alpha \cdot \mathbf{G}_{\mathcal{C}}^t + (1 - \alpha) \cdot \mathbf{G}_{\mathcal{U}}^t. \tag{7}$$

Then we combine the subspace fusion with cross-enhancement, ensuring training stability through residual connection and layer normalization:

$$\mathbf{P}^{t+1} = \text{LayerNorm}\left( \text{FC}\left( \tilde{\mathbf{P}}^{t+1} + \mathbf{P}_{cross}^t \right) + \mathbf{P}^t \right), \tag{8}$$

where FC is a fully-connected layer and the residual connection $+\mathbf{P}^t$ prevents gradient vanishing. Finally, we apply L2 normalization: $\mathbf{P}^{t+1} \leftarrow \mathbf{P}^{t+1}/(\|\mathbf{P}^{t+1}\|_2 + \epsilon)$ with $\epsilon = 10^{-8}$ to maintain geometric properties.

### 3.4. Progressive Refinement and Mixture of Prototype Experts

Single-step refinement causes the prototype to jump directly from $\mathbf{P}^0$ to $\mathbf{P}^T$, easily leading to over-adaptation or catastrophic forgetting. Progressive refinement establishes a controlled trajectory through multiple cascaded steps. By iteratively applying the DPR module $T$ times, we construct an evolution trajectory $P_{Traj} = \{\mathbf{P}^0, \mathbf{P}^1, \ldots, \mathbf{P}^T\}$ from support-biased to query-adapted representations.

At each step $t$, we generate intermediate predictions through prototype matching:

$$\mathbf{z}_i^t = \text{softmax}\left( \frac{\mathbf{f}_i^q \cdot (\mathbf{P}^t)^\top}{\sqrt{D}} \right), \tag{9}$$

where $\mathbf{f}_i^q$ is the feature of query point $i$, and $\mathbf{z}_i^t \in \mathbb{R}^{N+1}$ is its prediction distribution over $N + 1$ classes.

**MoPE Mechanism.** To fully leverage the diversity along the refinement trajectory, we treat each prototype $\mathbf{P}^t$ as an "expert" at different refinement stages: early experts ($t \approx 0$) retain more support-biased features with strong discriminative power; middle experts ($t \approx T/2$) balance both distributions; late experts ($t \approx T$) are highly adapted to the query set.

MoPE adopts a confidence-driven weighted ensemble strategy. For each query point $i$ and expert $t$, we compute the prediction confidence $c_i^t = \max(\mathbf{z}_i^t)$. These confidences are converted into mixing weights through softmax normalization:

$$w_i^t = \frac{\exp(c_i^t/\tau_{\text{mix}})}{\sum_{t'=0}^{T} \exp(c_i^{t'}/\tau_{\text{mix}})}, \tag{10}$$

where temperature parameter $\tau_{\text{mix}} = 1.0$ controls weight distribution sharpness. The final prediction is obtained through weighted ensemble:

$$\hat{\mathbf{z}}_i = \sum_{t=0}^{T} w_i^t \cdot \mathbf{z}_i^t. \tag{11}$$

This mechanism allows different query points to adaptively select the most suitable expert combination based on their feature distributions.

### 3.5. Training Objectives

The total training objective function $\mathcal{L}$ combines the standard cross-entropy loss $\mathcal{L}_{\text{CE}}$ with three auxiliary regularization terms:

$$\mathcal{L} = \mathcal{L}_{CE} + \lambda_c \mathcal{L}_c + \lambda_{ic} \mathcal{L}_{ic} + \lambda_{ig} \mathcal{L}_{ig}, \tag{12}$$

where all auxiliary loss weights are set to $\lambda_{\text{c}} = \lambda_{\text{ic}} = \lambda_{\text{ig}} = 0.1$.

**Classification Loss.** The standard cross-entropy loss supervises the final MoPE prediction:

$$\mathcal{L}_{\text{CE}} = -\frac{1}{M_q} \sum_{i=1}^{M_q} \sum_{c=0}^{N} y_i^c \log \hat{z}_i^c, \tag{13}$$

where $y_i^c$ is the one-hot ground truth label for point $i$.

**Channel Correlation Loss.** This loss ensures that the refinement direction is highly correlated with query feature activation patterns:

$$\mathcal{L}_{\text{c}} = \frac{1}{T} \sum_{t=1}^{T} \left( 1 - \frac{\mathbf{P}^t \cdot \bar{\mathbf{F}}^q}{\|\mathbf{P}^t\|\|\bar{\mathbf{F}}^q\|} \right), \tag{14}$$

where $\bar{\mathbf{F}}^q$ represents the mean query features.

**Inter-Class Diversity Loss.** This loss enhances the discriminative power of the final prototype $\mathbf{P}^T$:

$$\mathcal{L}_{\text{ic}} = \sum_{c \neq c'} \max(0, \delta - \|\mathbf{p}_c^T - \mathbf{p}_{c'}^T\|_2), \tag{15}$$

where $\delta = 0.5$ is the minimum margin threshold.

**Inter-Generation Diversity Loss.** This loss constrains the magnitude of changes between adjacent steps to approach target step size $\rho = 0.1$:

$$\mathcal{L}_{\text{ig}} = \frac{1}{T} \sum_{t=1}^{T} \left| \|\mathbf{P}^t - \mathbf{P}^{t-1}\|_2 - \rho \right|. \tag{16}$$

This ensures substantial progress at each refinement step.

## 4. Experiments

### 4.1. Datasets and Evaluation Metrics

**S3DIS dataset** (Armeni et al., 2016) comprises RGB-colored point clouds captured from 272 indoor rooms spanning 6 distinct buildings (conferencing, auditoriums, offices,

*Table 1.* **Quantitative results on S3DIS dataset.** We report mIoU (%) for different N-way K-shot settings. Best results are highlighted in **bold**. $\Delta$ Improvement is reported relative to the Seg-PN (raw coords, w/o RGB) baseline.

| Method | 2-Way | | | | | | 3-Way | | | | | |
|---|---|---|---|---|---|---|---|---|---|---|---|---|
| | 1-Shot | | | 5-Shot | | | 1-Shot | | | 5-Shot | | |
| | $S_0$ | $S_1$ | Mean | $S_0$ | $S_1$ | Mean | $S_0$ | $S_1$ | Mean | $S_0$ | $S_1$ | Mean |
| DGCNN (Wang et al., 2019) | 36.34 | 38.79 | 37.57 | 56.49 | 56.99 | 56.74 | 30.05 | 32.19 | 31.12 | 46.88 | 47.57 | 47.23 |
| ProtoNet (Snell et al., 2017) | 48.39 | 49.98 | 49.19 | 57.34 | 63.22 | 60.28 | 40.81 | 45.07 | 42.94 | 49.05 | 53.42 | 51.24 |
| MPTI (Zhao et al., 2021b) | 52.27 | 51.48 | 51.88 | 58.93 | 60.56 | 59.75 | 44.27 | 46.92 | 45.60 | 51.74 | 48.57 | 50.16 |
| AttMPTI (Zhao et al., 2021b) | 53.77 | 55.94 | 54.86 | 61.67 | 67.02 | 64.35 | 45.18 | 49.27 | 47.23 | 54.92 | 56.79 | 55.86 |
| BFG (Mao et al., 2022) | 55.60 | 55.98 | 55.79 | 63.71 | 66.62 | 65.17 | 46.18 | 48.36 | 47.27 | 55.05 | 57.80 | 56.43 |
| 2CBR (Zhu et al., 2023) | 55.89 | 61.99 | 58.94 | 63.55 | 67.51 | 65.53 | 46.51 | 53.91 | 50.21 | 55.51 | 58.07 | 56.79 |
| PAP3D (He et al., 2023) | 59.45 | 66.08 | 62.76 | 65.40 | 70.30 | 67.85 | 48.99 | 56.57 | 52.78 | 61.27 | 60.81 | 61.04 |
| DPA (Liu et al., 2024) | 66.08 | 74.30 | 70.19 | 71.10 | 77.03 | 74.07 | 50.67 | 59.53 | 55.10 | 64.52 | 63.34 | 63.93 |
| TaylorSeg-PN (Wang et al., 2025c) | 67.12 | 71.11 | 69.12 | 70.44 | 72.23 | 71.34 | 60.28 | 65.70 | 63.00 | 62.78 | 67.06 | 64.33 |
| APR (Jiang et al., 2025) | 67.75 | 69.45 | 68.60 | 70.28 | 73.38 | 71.83 | 61.03 | 64.23 | 62.63 | 65.61 | 70.78 | 68.20 |
| DAFNet (Wang et al., 2025b) | 68.13 | 70.27 | 69.20 | 70.51 | 73.15 | 71.83 | 61.33 | 65.55 | 63.44 | 65.25 | 68.67 | 66.96 |
| DyPolySeg (Wang et al., 2025a) | 72.02 | 73.82 | 72.92 | 75.99 | 75.32 | 75.66 | 64.54 | 67.93 | 66.24 | 65.61 | 70.22 | 67.92 |
| Seg-PN (Zhu et al., 2024) (norm. coords + RGB) | 64.84 | 67.98 | 66.41 | 67.63 | 71.48 | 69.56 | 59.11 | 60.42 | 59.77 | 59.48 | 64.72 | 62.10 |
| Seg-PN (Zhu et al., 2024) (raw coords, w/o RGB) | 77.11 | 76.78 | 76.95 | 81.14 | 77.57 | 79.35 | 68.73 | 73.85 | 71.29 | 75.24 | **73.81** | 74.52 |
| **DPR-Net (Ours)** | **80.76** | **77.78** | **79.27** | **81.27** | **79.92** | **80.60** | **72.94** | **74.61** | **73.77** | **75.46** | 73.65 | **74.55** |
| $\Delta$ *Improvement* | +3.65 | +1.00 | +2.32 | +0.13 | +2.35 | +1.25 | +4.21 | +0.76 | +2.48 | +0.22 | -0.16 | +0.03 |

*Table 2.* Seen and Unseen Classes Split for S3DIS and ScanNet. We follow (Zhao et al., 2021b) to evenly assign categories to $S_0$ and $S_1$ splits.

| | $S_0$ | $S_1$ |
|---|---|---|
| S3DIS | beam, board, bookcase, ceiling, chair, column | door, floor, sofa, table, wall, window |
| ScanNet | bathtub, bed, bookshelf, cabinet, chair, counter, curtain, desk, door, floor | other furniture, picture, refrigerator, show curtain, sink, sofa, table, toilet, wall, window |

etc.). Each point is annotated with one of 13 semantic categories, including 12 object classes plus a clutter category for ambiguous regions.

**ScanNet dataset** (Dai et al., 2017) contains 1513 densely-scanned indoor scenes with high-quality point-wise semantic annotations. It covers 20 semantic categories, excluding unannotated background regions.

**Evaluation Metric:** We employ mean Intersection over Union (mIoU) as the performance evaluation metric.

## 4.2. Implementation Details

Following the preprocessing protocol established by (Zhao et al., 2021b), we partition each scene into 1m × 1m spatial blocks to maintain local geometric coherence. From each block, we uniformly sample 2048 points with their corresponding 3D coordinates and RGB features. To simulate realistic few-shot learning scenarios, we partition the semantic categories of both datasets into two non-overlapping subsets: $S_0$ (seen classes for training) and $S_1$ (unseen classes for testing). As detailed in Table 2, S3DIS categories are evenly

divided into two splits of 6 classes each, while ScanNet categories are split into two subsets of 10 classes each.

All experiments are conducted on a single NVIDIA GeForce RTX 5090 GPU with 32GB memory using PyTorch 2.8 and CUDA 13.0. We adopt episodic learning under N-way K-shot settings where $N \in \{2, 3\}$ and $K \in \{1, 5\}$. For each configuration, we sample 100 test episodes and report average mIoU. Point clouds are voxelized and uniformly sampled to 2048 points. We found that using normalized point coordinates and RGB information would significantly degrade model performance. Therefore, in this paper, we only used the absolute coordinates of the points. We adopt Seg-NN (Zhu et al., 2024) as our feature extraction backbone. During training, we apply data augmentation including random rotation, translation, and jitter. Model parameters are optimized using AdamW optimizer ($\beta_1 = 0.9$, $\beta_2 = 0.999$) with initial learning rate 0.001.

## 4.3. Comparison with State-of-the-Art Methods

### 4.3.1. RESULTS ON S3DIS DATASET

A critical yet overlooked bottleneck of prior FS-PCSS methods lies in the input representation: they conventionally feed block-normalized coordinates together with RGB. As shown in Table 1, keeping the Seg-PN encoder unchanged and merely replacing normalized coordinates and RGB with *raw absolute* coordinates raises the mean mIoU by 10.5–12.4 points across all settings (e.g., $66.41 \rightarrow 76.95$ in 2-way 1-shot), as normalization discards discriminative scale and position cues while RGB injects appearance domain shift between support and query. Remarkably, the plain Seg-PN baseline under absolute coordinates already surpasses all prior normalized-protocol methods, exceeding the strongest

*Table 3.* **Quantitative results on ScanNet dataset. We report mIoU (%) for different N-way K-shot settings.** Best results are highlighted in **bold**. $\Delta$ Improvement is reported relative to the Seg-PN (raw coords, w/o RGB) baseline.

| Method | 2-Way | | | | | | 3-Way | | | | | |
|---|---|---|---|---|---|---|---|---|---|---|---|---|
| | 1-Shot | | | 5-Shot | | | 1-Shot | | | 5-Shot | | |
| | $S_0$ | $S_1$ | Mean | $S_0$ | $S_1$ | Mean | $S_0$ | $S_1$ | Mean | $S_0$ | $S_1$ | Mean |
| DGCNN (Wang et al., 2019) | 31.55 | 28.94 | 30.25 | 42.71 | 37.24 | 39.98 | 23.99 | 19.10 | 21.55 | 34.93 | 28.10 | 31.52 |
| ProtoNet (Snell et al., 2017) | 33.92 | 30.95 | 32.44 | 45.34 | 42.01 | 43.68 | 28.47 | 26.13 | 27.30 | 37.36 | 34.98 | 36.17 |
| MPTI (Zhao et al., 2021b) | 39.27 | 36.14 | 37.71 | 46.90 | 43.59 | 45.25 | 29.96 | 27.26 | 28.61 | 38.14 | 34.36 | 36.25 |
| AttMPTI (Zhao et al., 2021b) | 42.55 | 40.83 | 41.69 | 54.00 | 50.32 | 52.16 | 35.23 | 30.72 | 32.98 | 46.74 | 40.80 | 43.77 |
| BFG (Mao et al., 2022) | 42.15 | 40.52 | 41.34 | 51.23 | 49.39 | 50.31 | 34.12 | 31.98 | 33.05 | 46.25 | 41.38 | 43.82 |
| 2CBR (Zhu et al., 2023) | 50.73 | 47.66 | 49.20 | 52.35 | 47.14 | 49.75 | 47.00 | 46.36 | 46.68 | 45.06 | 39.47 | 42.27 |
| PAP3D (He et al., 2023) | 57.08 | 55.94 | 56.51 | 64.55 | 59.64 | 62.10 | 55.27 | 55.60 | 55.44 | 59.02 | 53.16 | 56.09 |
| DPA (Liu et al., 2024) | 62.75 | 63.04 | 62.90 | 67.19 | 64.62 | 65.91 | 61.97 | 61.72 | 61.85 | 66.13 | 64.67 | 65.40 |
| APR (Jiang et al., 2025) | 67.07 | 67.31 | 67.19 | 68.61 | 69.72 | 69.17 | 63.74 | 65.78 | 64.76 | 65.35 | 68.37 | 66.86 |
| TaylorSeg-PN (Wang et al., 2025c) | 67.52 | 70.75 | 69.14 | 68.39 | 71.55 | 69.97 | 63.60 | 67.55 | 65.58 | 66.98 | 69.78 | 68.38 |
| DAFNet (Wang et al., 2025b) | 68.79 | 69.95 | 69.37 | 70.91 | 70.60 | 70.76 | 66.14 | 66.70 | 66.42 | 68.97 | 71.95 | 70.46 |
| DyPolySeg (Wang et al., 2025a) | 71.05 | **72.73** | **71.89** | 71.25 | 73.66 | 72.46 | 67.65 | 71.24 | 69.45 | 68.73 | 69.62 | 69.18 |
| Seg-PN (Zhu et al., 2024) (norm. coords + RGB) | 63.15 | 64.32 | 63.74 | 67.08 | 69.05 | 68.07 | 61.80 | 65.34 | 63.57 | 62.94 | 68.26 | 65.60 |
| Seg-PN (Zhu et al., 2024) (raw coords, w/o RGB) | 69.25 | 64.49 | 66.87 | **74.10** | 73.72 | 73.91 | 67.47 | 70.22 | 68.84 | 72.71 | 72.55 | 72.63 |
| **DPR-Net (Ours)** | **71.44** | 71.70 | 71.57 | 73.95 | **74.51** | **74.23** | **69.75** | **72.52** | **71.14** | **73.75** | **72.85** | **73.30** |
| $\Delta$ *Improvement* | +2.19 | +7.21 | +4.70 | -0.15 | +0.79 | +0.32 | +2.28 | +2.30 | +2.30 | +1.04 | +0.30 | +0.67 |

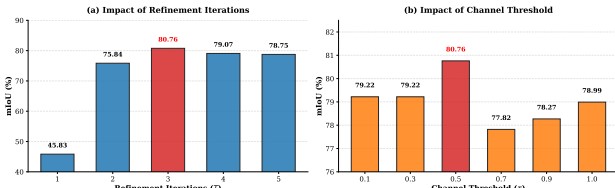

*Figure 3.* **Ablation studies on discrete hyperparameters.** (a) Impact of refinement iterations $T$ with optimal at $T = 3$ (80.76%). (b) Impact of channel threshold $\tau$ with optimal at $\tau = 0.5$. Red bars indicate optimal values.

competitor DyPolySeg (Wang et al., 2025a) by $+4.03$ mean mIoU in 2-way 1-shot. Building on this with the same lightweight encoder and only 0.04M extra parameters, DPR-Net attains the best results in 11 of 12 columns, adding a further $+2.32/+1.25/+2.48$ mean mIoU in 2-way 1-shot/2-way 5-shot/3-way 1-shot over the fair absolute-coordinate baseline, with gains saturating only in the supervision-rich 3-way 5-shot setting.

### 4.3.2. Results on ScanNet Dataset

As shown in Table 3, the same trend holds on ScanNet: switching the Seg-PN baseline from normalized coordinates with RGB to raw absolute coordinates lifts the mean mIoU by 3.1–7.0 points (e.g., $65.60 \rightarrow 72.63$ in 3-way 5-shot), confirming that the input representation is a dataset-agnostic bottleneck. Built on the same encoder, DPR-Net further improves over this strong absolute-coordinate baseline by $+4.70/+0.32/+2.30/+0.67$ mean mIoU across the four settings and ranks first in 9 of the 12 columns. It exceeds the strongest prior method DyPolySeg (Wang et al., 2025a) by $+1.77$, $+1.69$, and $+4.12$ mean mIoU in 2-way 5-shot, 3-way 1-shot, and 3-way 5-shot, and remains on par in 2-

way 1-shot (leading on $S_0$ while trailing slightly on $S_1$), demonstrating that the dual-subspace refinement and MoPE ensembling generalize to complex, category-rich indoor scenes.

### 4.4. Ablation Studies

All the experiments below were conducted on a backbone with Seg-NN (Zhu et al., 2024) as the encoder.

#### 4.4.1. Impact of Refinement Iterations

As shown in Fig. 3(a), $T$ significantly affects performance. Starting from $T = 1$ with 45.83% mIoU, performance dramatically improves to 75.84% at $T = 2$, demonstrating the necessity of iterative refinement over single-step adaptation. Performance peaks at $T = 3$ with 80.76% mIoU, our adopted configuration. However, further iterations cause degradation: $T = 4$ yields 79.07% and $T = 5$ achieves 78.75%, revealing that excessive iterations cause over-adaptation. The optimal $T = 3$ balances adaptation and feature preservation, achieving substantial improvement while avoiding performance degradation at $T \geq 4$.

#### 4.4.2. Impact of Channel Activation Threshold

The threshold $\tau$ serves as a critical gating mechanism for dual-subspace decomposition. As shown in Fig. 3(b), $\tau$ exhibits clear sensitivity with optimal performance at 0.5 (80.76% mIoU). Lower thresholds ($\tau = 0.1, 0.3$) yield 79.22%, as overly permissive criteria admit noisy channels that dilute discriminative information. Higher thresholds degrade performance ($\tau = 0.7$: 77.82%, $\tau = 0.9$: 78.27%), as strict criteria exclude discriminative channels essential for accurate segmentation. The optimal $\tau = 0.5$ balances inclu-

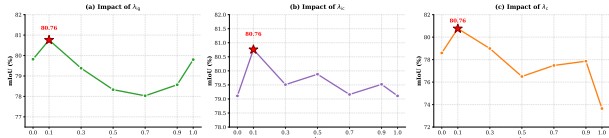

*Figure 4.* **Sensitivity analysis of auxiliary loss weight coefficients.** (a) $\lambda_{ig}$ with optimal at 0.1. (b) $\lambda_{ic}$ demonstrates robustness with optimal at 0.1. (c) $\lambda_c$ shows strong sensitivity with degradation beyond 0.1. Red stars indicate optimal values.

siveness and selectivity, achieving consistent improvements over both permissive and overly strict criteria.

#### 4.4.3. EFFECTIVENESS OF DPR MODULE COMPONENTS

Table 4 presents ablation results by systematically removing components. Removing spatial features causes the most dramatic drop to 47.31% ($-33.45$ points), revealing that cross-correlation attention is indispensable for spatial correspondences. Ablating common subspace features reduces performance to 78.56% ($-2.20$ points), indicating its crucial role in semantic preservation. Removing dual-subspace decomposition yields 78.99% ($-1.77$ points). These results validate that spatial features are paramount, followed by common subspace and distinctive features.

#### 4.4.4. IMPACT OF AUXILIARY LOSS FUNCTIONS

Table 5 shows results of progressively adding auxiliary losses. Using only $\mathcal{L}_{CE}$ achieves 78.62% mIoU as the baseline. Adding channel correlation loss $\mathcal{L}_c$ improves performance to 78.95% (+0.32 points), demonstrating its role in guiding refinement toward query-relevant features. Further incorporating inter-class diversity loss $\mathcal{L}_{ic}$ yields 79.78% (+1.16 points), showing that maintaining class separation enhances discriminative power. Incorporating all three losses achieves optimal 80.76% (+2.14 points), with inter-generation diversity loss $\mathcal{L}_{ig}$ contributing an additional +0.98 points by ensuring consistent progress across refinement steps. The cumulative improvement of 2.14 points validates the synergistic effect of all auxiliary losses in stabilizing training and enhancing prototype quality throughout the refinement trajectory.

*Table 4.* Ablation study on DPR module components.

| Configuration | mIoU (%) | $\Delta$ |
|---|---|---|
| w/o Spatial Features | 47.31 | -33.45 |
| w/o Common Subspace | 78.56 | -2.20 |
| w/o Dual-Subspace Decomposition | 78.99 | -1.77 |
| w/o Distinctive Subspace | 79.15 | -1.61 |
| **Full Model** | **80.76** | - |

*Table 5.* Ablation study on auxiliary loss functions.

| Loss Configuration | mIoU (%) | $\Delta$ |
|---|---|---|
| $\mathcal{L}_{CE}$ only | 78.62 | - |
| $\mathcal{L}_{CE} + \mathcal{L}_c$ | 78.95 | +0.32 |
| $\mathcal{L}_{CE} + \mathcal{L}_c + \mathcal{L}_{ic}$ | 79.78 | +1.16 |
| **Full Loss** (+ $\mathcal{L}_{ig}$) | **80.76** | **+2.14** |

### 4.5. Ablation Study on MoPE Voting Strategies

*Table 6.* **Ablation study on MoPE voting strategies.** All experiments are conducted on S3DIS S0 split under 2-way 1-shot setting.

| Voting Strategy | mIoU (%) | $\Delta$ from Best |
|---|---|---|
| Trajectory Smoothing | 76.56 | -4.20 |
| Simple Majority Voting | 76.97 | -3.79 |
| Learnable Confidence Weighting | 79.32 | -1.44 |
| Learnable Weights | 79.43 | -1.33 |
| **Confidence-based (Ours)** | **80.76** | **–** |

As shown in Table 6, we conduct comprehensive ablation experiments comparing five different voting strategies. This ablation validates MoPE's confidence-based weighting against four alternatives. Our method achieves 80.76% mIoU, outperforming simple majority voting (76.97%, +3.79 points) and learnable approaches (79.32-79.43%, +1.33-1.44 points). The substantial gaps demonstrate that adaptive per-point confidence provides more robust signals than uniform weighting or fixed learned parameters.

### 4.6. MoPE Weight Distribution Analysis

Fig. 5 visualizes the ensemble weight distribution of MoPE. As shown in Fig. 5(a), the mean weights assigned to prototypes $P^0$, $P^1$, and $P^2$ are 0.264, 0.342, and 0.393, respectively. The monotonically increasing trend confirms that later-stage prototypes, being more adapted to the query distribution, receive higher confidence. Notably, the early prototype $P^0$ still retains a non-trivial weight of 0.264, indicating that support-biased semantics remain valuable and are not discarded. Fig. 5(b) presents a per-point weight heatmap, where each row corresponds to a query point and each column to a prototype expert. The heterogeneous pattern across rows demonstrates that different query points rely on different refinement stages, validating MoPE's confidence-driven, point-wise ensemble mechanism.

#### 4.6.1. SENSITIVITY ANALYSIS OF LOSS WEIGHT COEFFICIENTS

Fig. 4 visualizes sensitivity analysis of auxiliary loss coefficients. $\lambda_{ig}$ exhibits moderate sensitivity with optimal at 0.1 (80.76%). $\lambda_{ic}$ demonstrates robustness across 0.3-0.9 range, with optimal at 0.1 (80.76%). $\lambda_c$ exhibits strongest sensitiv-

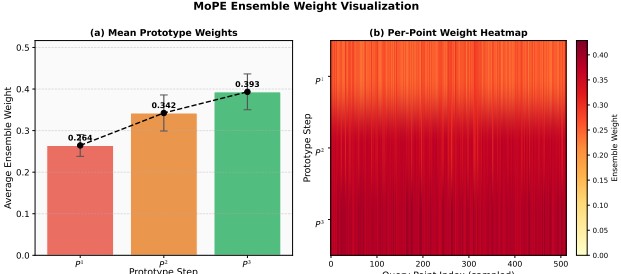

*Figure 5.* **MoPE weight distribution.** (a) Mean weights per prototype. (b) Per-point weight heatmap showing adaptive expert selection.

ity, with optimal at 0.1 (80.76%). Performance deteriorates sharply as it increases: $\lambda_c = 0.3$ yields 78.99%, $\lambda_c = 0.5$ drops to 76.50%, and $\lambda_c = 1.0$ crashes to 73.65%, revealing that over-emphasizing channel alignment constrains adaptive flexibility.

### 4.6.2. MODEL COMPLEXITY ANALYSIS

Table 7 compares computational efficiency under 2-way 1-shot on S3DIS $S_0$. DPR-Net achieves 80.76% mIoU with 0.28M parameters and 8.37 GFLOPs. Compared to Seg-PN (Zhu et al., 2024) under the original normalized-coordinate and RGB setting (64.84%), DPR-Net improves performance by 15.92 points, while still requiring only 0.04M additional parameters. DPR-Net significantly outperforms heavier models like DPA (Liu et al., 2024) (5.08M, 15.67G, 66.08%) while using only 5.5% of its parameters and 53.4% of its FLOPs.

*Table 7.* Model complexity comparison on $S_0$ split of S3DIS under 2-way 1-shot setting.

| Method | Params (M) | FLOPs (G) | mIoU (%) |
| --- | --- | --- | --- |
| AttMPTI (Zhao et al., 2021b) | 0.36 | 152.65 | 53.77 |
| QGPA (Hu et al., 2023) | 2.79 | 16.30 | 56.30 |
| PAP3D (He et al., 2023) | 2.57 | 15.05 | 59.45 |
| DPA (Liu et al., 2024) | 5.08 | 15.67 | 66.08 |
| Seg-PN (Zhu et al., 2024) | 0.24 | 8.36 | 64.84 |
| **DPR-Net (Ours)** | 0.28 | 8.37 | **80.76** |

### 4.7. Limitations

While DPR-Net achieves state-of-the-art performance on indoor scene datasets, we acknowledge several limitations. First, our method is primarily evaluated on indoor datasets (S3DIS, ScanNet), and generalization to outdoor scenes (e.g., autonomous driving scenarios) remains to be validated. Second, although lightweight (0.28M parameters), the iterative refinement process requires multiple forward passes, increasing inference time compared to single-pass methods. Third, the optimal number of refinement steps

($T = 3$) is dataset-dependent and may require tuning for new domains.

## 5. Conclusion

In this paper, we propose DPR-Net, a Deep Prototype Refinement Network for few-shot point cloud semantic segmentation. Our core DPR module decomposes features into common and distinctive subspaces via channel activation, enabling targeted adaptation while preserving shared semantics. By cascading multiple DPR modules, we construct a progressive refinement trajectory from support-biased to query-adapted representations, mitigating both under-adaptation and over-adaptation. The MoPE mechanism further ensembles intermediate prototypes through confidence-driven weighting, leveraging complementary information across refinement stages. Future work will extend our framework to outdoor point cloud scenes and explore multi-modal fusion for enhanced 3D understanding.

## Impact Statement

This paper advances few-shot 3D point cloud semantic segmentation, enabling accurate scene understanding from only a handful of labeled examples. By lowering the costly annotation burden of 3D data, our work can broaden access to 3D perception for applications in robotics, augmented reality, and autonomous systems. As with most perception technologies, these capabilities are dual-use: when deployed without safeguards, 3D scene segmentation could facilitate unauthorized surveillance or reveal private environments, a concern highlighted by the indoor scenes (e.g., S3DIS, Scan-Net) used in our benchmarks. Our experiments rely solely on established public datasets, and we do not believe this work introduces risks beyond those inherent to 3D perception research. We encourage responsible deployment with appropriate privacy protections and regulatory compliance.

## Acknowledgments

This work was supported in part by the European Union's Horizon 2024 Research and Innovation Programme for the Marie Skłodowska-Curie Actions under Grant No. 101211118; the Scientific and Technological Innovation Project of China Academy of Chinese Medical Sciences under Grant CI2023C001YG; and the UKRI Future Leaders Fellowship under Grant MR/V025333/1 (RoboHike). Shuting He was sponsored by Shanghai Pujiang Programme 24PJD030 and Natural Science Foundation of Shanghai 25ZR1402138. Xingyu Gao was sponsored by Brain Science and Brain-like Intelligence Technology— National Science and Technology Major Project under Grant 2022ZD0208700, and National Natural Science Foundation of China under Grant 62376264.

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
