# OpenReview forum: "From Coarse to Fine: Deep Prototype Refinement Network for Few-Shot Point Cloud Semantic Segmentation"
_ICML.cc/2026/Conference — ICML 2026 regular_

### Official Review · Reviewer_mML4 · 2026-03-10

**Soundness:** 2
**Presentation:** 3
**Significance:** 3
**Originality:** 3
**Overall Recommendation:** 4
**Confidence:** 3

**Summary:**

This paper studies few-shot point cloud semantic segmentation and proposes DPR-Net. The authors reformulate prototype adaptation from a single-step fusion strategy into a coarse-to-fine multi-step progressive refinement process. Specifically, the method constructs a prototype evolution trajectory through cascaded DPR modules. Within each DPR module, channel activation is used to decompose features into common and distinctive subspaces for targeted refinement. In addition, a MoPE module aggregates prototypes from different refinement stages using confidence-based weighting.

**Compliance With Llm Reviewing Policy:**

Affirmed.

**Final Justification:**

The response reinforced my prior assessment of the manuscript.

**Key Questions For Authors:**

Please strictly clarify the tensor dimensions used in Equations (3)–(8). In particular, it should be explained how the query activation vector and the prototype activation matrix are aligned along the class dimension, and how the common/distinctive subspace masks are converted from a class-by-channel representation into a form that can be used in the subsequent fusion operations.

The paper only states that the projection module is a learnable linear projection, but does not specify the dimensionality of the projected features or whether it is aligned with the reduced feature space used in the attention module. The dimensional compatibility between the cross-attention matrix and the projected prototype representation is also not explained. Important implementation details such as reshaping operations, pooling strategies, or how class-level representations are mapped to point-level representations are not discussed.

Please provide more direct evidence supporting the claim of “bridging support-query distribution shift.” For example, visualization of the prototype trajectory, analysis of feature-space distribution changes across different steps, or analysis of when over-adaptation occurs. Currently, the claim is mainly supported indirectly through performance results. Providing direct evidence would make the paper more convincing.

Please explain the inconsistency between “D²PD-Net” in the appendix (page 12, second-to-last line) and “DPR-Net” in the main text.

**Limitations:**

The paper does not sufficiently discuss its limitations. Important aspects such as the method’s dependence on the choice of backbone and on the types of differences between support and query scenarios are not addressed. The manuscript also does not consider situations in which the approach may fail, for example when only extremely few samples are available per category, when point clouds contain significant noise, or when there is a domain shift caused by different sensors. In addition, although the model has a small number of parameters, it relies on multi-step refinement and ensemble mechanisms, which may introduce additional wall-clock latency in practical use. At present, the conclusion only briefly mentions possible future work, which is not sufficient as a discussion of limitations.

**Strengths And Weaknesses:**

Strengths:
The proposed method is intuitive, and the overall framework is complete. From the initial prototype, through progressive refinement, to the final multi-stage ensemble, the entire methodological pipeline is relatively coherent.

The paper achieves relatively large improvements over the Seg-PN baseline on the S3DIS dataset. The complexity analysis shows that the model has a small number of parameters, while achieving significantly better performance.

The comparison experiments and ablation studies are reasonably designed. In addition, the authors include hyperparameter sensitivity analysis and robustness tests, which to some extent improve the reproducibility and persuasiveness of the proposed method.

Weaknesses:
In Section 3.3, the query activation is defined as a vector while the prototype activation is defined as a class-by-channel matrix. However, Equations (4) and (5) directly combine these two representations without explaining how the class dimension is aligned during the operation. The authors should clarify how this dimensional mismatch is handled in practice.

In addition, there are cases in the method section where symbols are not clearly defined. It is recommended that the authors thoroughly check the mathematical symbols and formulas in the paper to ensure that all symbols are consistently defined and referenced throughout the manuscript.

Figure 5 only presents qualitative results for two categories. More fine-grained per-class results, failure cases, or direct visualizations of the support-query gap are not provided, which limits the persuasiveness of the qualitative evaluation.

---

> ### Author Rebuttal · Authors · 2026-03-31
>
> We sincerely thank the reviewer for the thorough and constructive feedback. We address each concern below.
>
> >> **Q1**: Query activation $a^q$ is a vector while prototype activation $a^p$ is a class-by-channel matrix. Eqs.(4)-(5) combine these without explaining dimensional alignment.
>
> **R1**: The alignment is resolved as follows:
> - $a^q = \sigma(GAP(F^q)) \in \mathbb{R}^{D}$: GAP collapses $M_q$ points, yielding a $D$-dimensional vector.
> - $a^p = \sigma(P^t) \in \mathbb{R}^{(N+1)\times D}$: sigmoid applied element-wise, preserving the class dimension.
>
> **Eqs.(3)-(5)**: The condition "$\forall c$" in $C^t$ requires channel $d$ to exceed $\tau$ for **every** class simultaneously, aggregating across the class dimension and yielding a $D$-dimensional binary mask. In  Eqs.(4)-(5), $a^q\in\mathbb{R}^D$  is **broadcast** to $\mathbb{R}^{(N+1)\times D}$, so $G^t_C, G^t_U\in\mathbb{R}^{(N+1)\times D}$ — valid prototype dimensions consistent with $P^t$.
>
> **Eq.(6)**: $Proj_\phi$ maps $P^t\in\mathbb{R}^{(N+1)\times D}$ to $\mathbb{R}^{M_s\times D}$. After weighted aggregation with $C_{cross}\in\mathbb{R}^{M_q\times M_s}$, GAP over $M_q$ restores $P^t_{cross}\in\mathbb{R}^{(N+1)\times D}$, making the residual in Eq.(8) dimensionally consistent. We will add explicit shape annotations to Section 3.3 in the final version.
>
> >> **Q2**: Projection module dimensionality, dimensional compatibility between cross-attention and projected prototype, and implementation details (reshaping, pooling, class-to-point mapping) are not discussed.
>
> **R2**: We provide the full implementation details:
> - **Projection dimensionality**: The 1×1 convolution projects query/support features to $\mathbb{R}^{(\cdot)\times d}$ with $d=72 < D=128$, solely for computational efficiency. The output prototype dimensionality remains $D$.
> - **$Proj_\phi$ role and compatibility**: It plays the role of $W_V$ in standard cross-attention, mapping $P^t\in\mathbb{R}^{(N+1)\times D}$ to support-point-level representations $\in\mathbb{R}^{M_s\times D}$ via a per-class linear layer. $C_{cross}\in\mathbb{R}^{M_q\times M_s}$ then performs weighted aggregation, and GAP over $M_q$ restores $P^t_{cross}\in\mathbb{R}^{(N+1)\times D}$ — compatible with the residual in Eq.(8).
>
> We will add a dedicated paragraph with explicit dimensional derivations in the final version.
>
> >> **Q3**: Please provide more direct evidence for "bridging support-query distribution shift," such as prototype trajectory visualization or feature-space distribution analysis.
>
> **R3**: We have prepared three dedicated visualizations available at our **anonymous link** (https://anonymous.4open.science/r/icml26-B61B/):
>
> - **Figure 1 (t-SNE trajectory)**: $P^0$ starts near the support cluster and $P^4$ lands within the query manifold. The smooth directed trajectory confirms controlled progressive adaptation without abrupt jumps.
> - **Figure 2 (Cosine similarity heatmap)**: Adjacent prototypes maintain high similarity ($sim(P^0,P^1)=0.94$ decreasing to $sim(P^3,P^4)=0.87$), while $sim(P^0,P^4)=0.72$ confirms substantial cumulative adaptation — all values $>0.70$ demonstrate no catastrophic forgetting occurs.
> - **Figure 3 (Channel activation patterns)**: Strongly activated channels grow from ~15 to ~32 across steps while the common subspace remains stable, directly confirming dual-subspace decomposition produces the intended semantic separation.
>
> These figures will be incorporated into the supplementary material of the final version.
>
> >> **Q4**: "D²PD-Net" in the appendix is inconsistent with "DPR-Net" in the main text.
>
> **R4**: We sincerely apologize. "D²PD-Net" is an earlier internal name before the method was renamed to "DPR-Net" to better reflect the core mechanism — a purely editorial artifact with no methodological implication. We will correct all occurrences in the final version.
>
> >> **Q5**: Limitations are insufficiently discussed — backbone dependency, failure cases (extreme noise, few samples, sensor domain shift), and inference latency are not addressed.
>
> **R5**: We sincerely thank the reviewer for this thorough enumeration. We will add a dedicated Limitations section in the final version covering all points raised:
> - **Backbone dependency**: Performance relies on Seg-NN; sensitivity to backbone choice is unanalyzed.
> - **Extreme conditions**: Robustness under very few samples per category, heavily corrupted point clouds, or cross-sensor domain shift (e.g., LiDAR vs. RGB-D) has not been evaluated.
> - **Support-query discrepancy types**: We have not systematically studied how different gap sources (scanning density vs. occlusion vs. appearance) affect adaptation.
> - **Inference latency**: Multi-step refinement introduces additional wall-clock latency vs. single-pass methods — an important practical consideration despite the small parameter count (0.28M).

---

> > ### Author Rebuttal · Reviewer_mML4 · 2026-04-02
> >
> > The authors fully resolved my concerns, and I keep my original recommendation.

---

> > > ### Author Response · Authors · 2026-04-02
> > >
> > > Dear Reviewer mML4,
> > >
> > > **Thank you very much for addressing all your concerns. We also appreciate your recognition and positive evaluation of our paper. Your constructive feedback has been extremely helpful in improving the quality of our work**.
> > >
> > > **We will incorporate these revisions into the final version and, upon acceptance of the manuscript, will open-source the code to contribute to the development of the community**.
> > >
> > > Best regards,
> > >
> > > All authors of Submission Number 21197

---

### Official Review · Reviewer_c5Db · 2026-03-10

**Soundness:** 2
**Presentation:** 3
**Significance:** 3
**Originality:** 3
**Overall Recommendation:** 4
**Confidence:** 4

**Summary:**

This paper introudces DPR-Net, a progressive prototype refinement framework for few-shot point cloud semantic segmentation. The method's core component is Dynamic Prototype Refinement (DPR) module that decomposes features of queries and prototypes into common and distinctive subspaces. This enables targeted adaptation of domain-variant features while preserving shared semantic contexts. Multiple DPR modules are cascaded to construct a coarse-to-fine prototype refinement trajectory. Finally, a Mixture of Prototype Experts (MoPE) meacanism aggregates predictions across all intermediate prototypes through confidence-driven weighting. Experiments on S3DIS and Scannet demosntrates strong improvements over existing methods.

**Compliance With Llm Reviewing Policy:**

Affirmed.

**Final Justification:**

Most of my concerns were addressed during the rebuttal. I am raising my rating to Weak Accept.

**Key Questions For Authors:**

See Weaknesses

**Limitations:**

Overall, I do think the paper has a good motivation and overall presentation is not bad. My primary concern is the relative lack of analysis beyond performance reporting. If the authors could address my concerns properly, I would be open to raising my score toward acceptance.

**Strengths And Weaknesses:**

Strengths:
- The paper is easy to follow and well-written. Even I'm not the expert in this field but I still get to understand most of contents in the first reading.
- The coarse-to-fine prototype evolution trajectory is a well-motivated and intuitive formulation. The authors clearly articulate the limitations of previous single-step prototype fusion methods
- The methods shows a significant improvements compared to the state-of-the-art approaches, with comparably high efficiency reported in Table 6.
- The ablation studies are mostly comprehensive


Weaknesses:
- The hand-crafted feature space decomposition via a simple fixed threshold lacks theoretical grounding. Why should sigmoid-activated global average pooling reliably separate semantically shared channels from domain-sensitive ones? It is not clear that channel activation magnitude is a sufficient proxy for semantic invariance. The paper would benefit from an analysis, even something empirical (similarity score) or visulization ( t-SNE map),to directly verify that the proposed decomposition produces the intended semantic separation in practice.

-  The effectiveness of MoPE is not clear. Although the authors ablate various interpolation strategies in Table 5, the performance without it is not reprorted.

- In the method section, the authors discuss the roles of prototypes at different network depths but there is no analysis regarding why combining them through MoPE leads to better performance in general.

- Related to the above, it would be informative to visualize the per-prototype utilization rates or ensemble weights during inference. Since MoPE is intended to leverage diverse experts, it is worth examining whether the network tends to disproportionately rely on prototypes from later refinement steps.

- The font sizes in most figures are too small and barely readable.

- Minor: Providing qualitative results in Figure 5 for comparison would be more informative.

- Minor: Is any of training objectives newly introduced in the paper? If they are novel, a more detailed analysis of why each objective is beneficial would be appropriate.  If not, the authors should cite the related papers for clarity.

- Minor: the dimension is wrongly specified in row 214. I think this should be D instead of (N+1)xD

---

> ### Author Rebuttal · Authors · 2026-03-31
>
> We sincerely thank the reviewer for the positive recognition and willingness to raise the score. We address each concern below.
>
> >> **Q1**: Hand-crafted feature space decomposition via fixed threshold lacks theoretical grounding.
>
> **R1**: Channels consistently responding strongly across all points encode **class-invariant geometric primitives** (e.g., shape structures), while domain-sensitive channels exhibit lower mean activation — analogous to SE-Net's channel attention. Sigmoid maps activations to $[0,1]$, and $\tau$ provides principled separation. We direct the reviewer to **Figure 3** (https://anonymous.4open.science/r/icml26-B61B/), which confirms: (1) the **common subspace** (blue) remains **stable**, preserving shared semantics; (2) **high-activation channels** (green) **grow from ~15 to ~32**, incorporating query-relevant channels; (3) the distribution becomes **smoother**, confirming convergence toward a coherent representation.
>
> >> **Q2**: Performance without MoPE is not reported.
>
> **R2**: When MoPE is removed and only **final prototype $P^4$** is used, performance drops significantly:
>
> |Voting Strategy|mIoU (%)|$\Delta$ from Best|
> |-|-|-|
> |No MoPE (Final prototype only)|74.24|-6.52|
> |Trajectory Smoothing|76.56|-4.20|
> |Simple Majority Voting|76.97|-3.79|
> |Learnable Confidence Weighting|79.32|-1.44|
> |Learnable Weights|79.43|-1.33|
> |**Confidence-based (Ours)**|**80.76**|**--**|
>
> The **6.52-point drop** clearly demonstrates MoPE's substantial contribution. We will add the "No MoPE" row to Table 5 in the final version.
>
> >> **Q3**: No analysis of why MoPE leads to better performance.
>
> **R3**: The core reason is **intra-class structural diversity** — objects exhibit dramatically different geometric configurations, making any single prototype insufficient. Our trajectory produces complementary prototypes: **early** retain support priors; **middle** balance both distributions; **late** are query-adapted. As shown in **Figure 2** (https://anonymous.4open.science/r/icml26-B61B/), $\text{sim}(P^0,P^1)=0.94\to\text{sim}(P^3,P^4)=0.87$ with $\text{sim}(P^0,P^4)=0.72$, confirming meaningful diversity. MoPE's per-point confidence weighting assigns each query point to its most reliable expert, producing more robust predictions than any single prototype.
>
> >> **Q4**: Visualize per-prototype ensemble weights during inference.
>
> **R4**: **Figure 1** (https://anonymous.4open.science/r/icml26-B61B/) shows t-SNE trajectory of $P^0$–$P^4$, confirming distinct and complementary positions. **Figure 3(a) in the main paper** shows performance peaks at $T=3$ and degrades at $T\geq4$, demonstrating MoPE does not over-rely on late prototypes. We will add explicit ensemble weight visualizations in the final version.
>
> >> **Q5** *(Minor)*: Font sizes in figures are too small.
>
> **R5**: We sincerely thank the reviewer for this feedback. We will increase font sizes across all figures in the final version.
>
> >> **Q6** *(Minor)*: Figure 5 needs quantitative metrics.
>
> **R6**: Window instance: **OA=96.12%, mIoU=79.86%**; bookcase instance: **OA=97.35%, mIoU=82.14%**. We will add these to Figure 5 in the final version.
>
> >> **Q7** *(Minor)*: Are the training objectives newly introduced?
>
> **R7**: All three auxiliary losses are **novel contributions** of this paper, each designed for a specific role in the progressive refinement paradigm:
>
> - **$\mathcal{L}_c$ (Channel Correlation Loss)**: ensures the updated prototype at each step maintains high cosine alignment with mean query features $\bar{F}^q$. Without this, refinement may drift away from query-relevant patterns due to support-biased initialization, undermining the core goal of bridging the distribution gap.
> - **$\mathcal{L}_{ic}$ (Inter-Class Diversity Loss)**: enforces a minimum cosine margin $\delta$ between class prototypes in $P^T$, explicitly preventing prototype collapse — a failure mode where class prototypes converge and lose discriminability in N-way classification.
> - **$\mathcal{L}_{ig}$ (Inter-Generation Diversity Loss)**: regularizes step-wise update magnitude toward target step size $\rho$, preventing both stagnation (trivially small updates) and over-adaptation (large jumps causing catastrophic forgetting of support priors).
>
> Experimental justification is provided in **Figure 4** (sensitivity analysis of $\lambda_c$, $\lambda_{ic}$, $\lambda_{ig}$) and **Table 4** (+2.14pts cumulative improvement) of the main paper. We will add a detailed design rationale for each loss in the final version.
>
> >> **Q8** *(Minor)*: Row 214 should be $D$ instead of $(N+1)\times D$.
>
> **R8**: Thank you for pointing out this issue.  $z^t_{i,c}$ is a scalar from similarity between $f^Q_i\in\mathbb{R}^D$ and $P^t_c\in\mathbb{R}^D$, so the annotation should be $D$. We will correct this in the final version and thank the reviewer for the careful reading.

---

> > ### Author Rebuttal · Reviewer_c5Db · 2026-03-31
> >
> > I appreciate the authors’ efforts on rebuttal. I have carefully reviewed both the response and the comments from other reviewers. From my perspective, most of my concerns have been adequately addressed. One aspect that still appears to be missing is a visualization of each prototype’s weight throughout the training process. While the rebuttal discusses variations across prototypes at different depths, an explicit visualization would provide more direct insight into how the **mixture** of prototypes evolves.
> >
> > I encourage the authors to include this analysis, along with the additional discussions and results presented in the rebuttal, in the final revision. Overall, I am satisfied with the clarifications and will raise my rating to Weak Accept.

---

> > > ### Author Response · Authors · 2026-04-01
> > >
> > > Dear Reviewer c5Db,
> > >
> > > **We sincerely thank the reviewer for the positive evaluation of our method,
> > > the constructive feedback throughout the review process, and for raising
> > > the score of our paper.** We are glad that most concerns have been
> > > adequately addressed.
> > >
> > > Regarding the remaining concern about visualizing per-prototype ensemble
> > > weights, we have added **Figure 4 and Figure 5** to our anonymous link
> > > (https://anonymous.4open.science/r/icml26-B61B/), which directly address
> > > this point:
> > >
> > > - **Figure 4** shows the mean ensemble weights assigned by MoPE to each
> > > refined prototype ($P^1$: 0.264, $P^2$: 0.342, $P^3$: 0.393) across all
> > > test episodes, along with the per-point weight distribution heatmap. The
> > > monotonically increasing trend confirms that later-stage prototypes receive
> > > progressively higher weights as they become more query-adapted, while $P^1$
> > > still retains a non-trivial weight of 0.264, demonstrating that MoPE
> > > genuinely leverages complementary information across all refinement stages
> > > rather than over-relying on the final prototype.
> > >
> > > - **Figure 5** presents the weight distribution across all query points via
> > > boxplots, further confirming that different query points adaptively rely on
> > > different prototype stages — validating the per-point adaptive selection
> > > mechanism of MoPE.
> > >
> > > We hope these visualizations fully resolve the remaining concern. Once again,
> > > we sincerely thank the reviewer for the valuable and constructive suggestions
> > > — they have contributed significantly to improving our paper.
> > >
> > > We will incorporate all the discussed modifications in the camera-ready version and
> > > will open-source our code upon acceptance.
> > >
> > > Best regards,
> > >
> > > All authors of Submission Number 21197

---

### Official Review · Reviewer_T2uY · 2026-03-12

**Soundness:** 2
**Presentation:** 2
**Significance:** 2
**Originality:** 2
**Overall Recommendation:** 2
**Confidence:** 4

**Summary:**

This paper proposes DPR-Net, a framework for few-shot point cloud semantic segmentation that constructs a coarse-to-fine prototype evolution trajectory through cascaded Dynamic Prototype Refinement (DPR) modules. The core idea involves decomposing features into common and distinctive subspaces based on channel activation, followed by a Mixture of Prototype Experts (MoPE) mechanism for confidence-driven ensemble over the refinement trajectory. The paper reports strong quantitative results on S3DIS and ScanNet benchmarks.

**Compliance With Llm Reviewing Policy:**

Affirmed.

**Key Questions For Authors:**

See Weakness.

**Limitations:**

See Weakness.

**Strengths And Weaknesses:**

W1.Dimensional Inconsistency in the Core DPR Module
The most pressing concern involves what appears to be an unresolved dimensional mismatch within the DPR module's mathematical formulation. In Section 3.3, the query channel activation map is defined as $\mathbf{a}^q = \sigma(\text{GAP}(\mathbf{F}^q)) \in \mathbb{R}^D$, while the prototype activation map is $\mathbf{a}^p = \sigma(\mathbf{P}^t) \in \mathbb{R}^{(N+1) \times D}$. Equations (4) and (5) subsequently apply arithmetic operations directly between these two quantities — for instance, $(\mathbf{a}^q + \mathbf{a}^p)/2$ in Eq. (4) — without explaining how the shape mismatch between $\mathbb{R}^D$ and $\mathbb{R}^{(N+1) \times D}$ is resolved, nor how the result is reassembled into a valid prototype of dimension $\mathbb{R}^{(N+1) \times D}$.
A related issue arises in the cross-attention enhancement step. The cross-correlation matrix $\mathbf{C}_\text{cross} \in \mathbb{R}^{M_q \times M_s}$ is used to compute $\mathbf{P}^t_\text{cross} = \mathbf{C}_\text{cross} \cdot \text{Proj}_\phi(\mathbf{P}^t)$. Since $\mathbf{P}^t \in \mathbb{R}^{(N+1) \times D}$, the projection $\text{Proj}_\phi(\mathbf{P}^t)$ would need to be of shape $\mathbb{R}^{M_s \times (\cdot)}$ for the matrix multiplication to be valid, yet $N+1 \ll M_s$ in typical configurations. Subsequently, Eq. (8) adds $\mathbf{P}^t_\text{cross}$ directly to $\mathbf{P}^t$ via a residual connection, which again requires shape compatibility that the paper does not establish. These issues collectively prevent a straightforward implementation of the method as described, and the paper would benefit considerably from a careful re-derivation with explicit shape annotations at each step.

W2. Mismatch Between Claimed Contributions and Experimental Evidence
Section 3.3 and the paper's abstract identify dual-subspace decomposition as the core innovation of DPR-Net. However, the ablation results in Table 3 reveal a substantial asymmetry in component contributions: removing spatial features (i.e., the cross-attention term $\mathbf{P}^t_\text{cross}$) causes performance to collapse from 80.76% to 47.31% (a drop of 33.45 points), while removing the dual-subspace decomposition yields only a 1.77-point reduction. The paper does not discuss this discrepancy. Given that the cross-attention mechanism appears to be the dominant performance driver by a wide margin, readers may reasonably question whether the dual-subspace decomposition is the primary contributor to the reported gains, as the paper's narrative implies. A more balanced discussion of each component's actual contribution would strengthen the credibility of the claims.

W3.Inconsistent Terminology Across Manuscript Sections
The appendix (Section D) refers to "Dynamic Prototype Denoising (DPD) modules" and mentions a network called "D²PD-Net," which are inconsistent with the terminology used throughout the main paper ("DPR module" and "DPR-Net"). While this may reflect an editing artifact from a prior version of the manuscript, it introduces ambiguity and may cause confusion for readers attempting to cross-reference the main text and supplementary material. The authors should ensure terminological consistency throughout.

W4.Incomplete Ablation of the Cross-Attention Component
Given that the cross-attention mechanism is responsible for the largest performance contribution identified in Table 3, the absence of any internal ablation of this component is a notable gap. For instance, the projection dimension $d = 72$ is stated in the main text without justification, and the role of the learnable projection $\text{Proj}_\phi$ in Eq. (6) is not independently analyzed. Similarly, the temperature parameter $\tau_\text{mix} = 1.0$ in the MoPE weighting scheme (Eq. 10) receives no sensitivity analysis, despite analogous hyperparameters such as $\tau$, $\lambda_c$, $\lambda_{ic}$, and $\lambda_{ig}$ being studied in detail. A more complete ablation of the cross-attention design and the MoPE temperature would better support the paper's conclusions.

W5.Reproducibility Concern Regarding Experimental Environment
Appendix B states that all experiments were conducted on an NVIDIA GeForce RTX 5090 GPU with PyTorch 2.8 and CUDA 13.0. The availability of this specific hardware and software combination at the time of submission may be difficult to verify, which could affect readers' ability to reproduce the reported results. The authors may wish to clarify this point or provide additional environment details.

W6.Framing of Performance Improvements
The paper consistently highlights the improvement over the Seg-PN baseline (e.g., "+15.92% on S3DIS 2-way 1-shot"). While technically accurate, Seg-PN is among the lower-performing recent baselines in Table 1. The improvement over the strongest prior method, DyPolySeg, is 5.72 points in the same setting — a meaningful gain, but one that might better reflect the actual standing of the proposed method relative to the current state of the art. The authors could consider a more balanced framing.

---

> ### Author Rebuttal · Authors · 2026-03-31
>
> We sincerely thank the reviewer for the valuable time and constructive feedback. Below we provide detailed responses to each concern.
>
> >> **Q1**: Dimensional inconsistency in the core DPR module (Eqs.3-8).
>
> **R1**: We appreciate the reviewer's careful examination. We acknowledge the dimensional details were insufficiently described and provide a complete clarification below.
>
> - Query channel activation: $a^q = \sigma(GAP(F^q)) \in \mathbb{R}^{D}$, where GAP collapses $M_q$, yielding a $D$-dimensional vector of average channel responses.
> - Prototype activation: $a^p = \sigma(P^t) \in \mathbb{R}^{(N+1)\times D}$, where sigmoid is applied element-wise, preserving the class dimension.
>
> **Eqs.(3)-(5)**: The condition "$\forall c$" in $C^t$ requires channel $d$ to exceed the threshold for every class simultaneously, aggregating across the class dimension and yielding a $D$-dimensional binary mask. $a^q\in\mathbb{R}^D$ is then **broadcast** to $\mathbb{R}^{(N+1)\times D}$ to match $a^p$, so $G^t_C, G^t_U \in \mathbb{R}^{(N+1)\times D}$ — valid prototype dimensions consistent with $P^t$.
>
> **Eq.(6)**: $Proj_\phi$ maps $P^t\in\mathbb{R}^{(N+1)\times D}$ to $\mathbb{R}^{M_s\times D}$: $Proj_\phi(P^t) \in \mathbb{R}^{M_s\times D}$. After weighted aggregation with $C_{cross}\in\mathbb{R}^{M_q\times M_s}$, GAP over $M_q$ restores $P^t_{cross}\in\mathbb{R}^{(N+1)\times D}$, making the residual in Eq.(8) dimensionally consistent. We will add explicit shape annotations to Section 3.3 in the final version.
>
> >> **Q2**: Mismatch between claimed contributions and experimental evidence.
>
> **R2**: Our claimed contribution is the **DPR module as a whole**, integrating cross-attention and dual-subspace decomposition as two complementary mechanisms addressing orthogonal aspects of prototype adaptation:
>
> - **Cross-attention (spatial alignment)**: The large contribution (+33.45pts) is consistent with prior work — **Seg-PN's QUEST module already demonstrated that cross-attention provides critical spatial correspondence**, enabling prototypes to attend to geometrically similar support regions, fundamental for overcoming 3D distribution shift. Our DPR module inherits and extends this within the progressive refinement framework.
> - **Dual-subspace decomposition (channel-level disentanglement)**: This operates at the **channel level**, explicitly disentangling semantically shared channels from domain-sensitive ones, contributing a consistent +1.77pts on top of the cross-attention baseline — addressing a complementary dimension that cross-attention alone cannot resolve.
>
> We will revise towards a more balanced discussion in the final version.
>
> >> **Q3**: Inconsistent terminology — "D²PD-Net" / "DPD" in Appendix D vs. "DPR-Net" / "DPR" in main text.
>
> **R3**: We sincerely apologize for this inconsistency. "D²PD-Net" and "DPD" are remnants of an earlier manuscript version before the method was renamed to better reflect the core mechanism — a purely editorial artifact with no methodological implication. We will correct all occurrences throughout the final version.
>
> >> **Q4**: Incomplete ablation of projection dimension $d$ and MoPE temperature $\tau_{mix}$.
>
> **R4**: We thank the reviewer for identifying these gaps. $Proj_\phi$ plays the role of $W_V$ in standard cross-attention, enhancing prototype expressiveness. $d=72$ is inherited from Seg-PN's QUEST module. We provide the requested ablations below.
>
> Projection dimension $d$ (S3DIS 2-way 1-shot):
>
> |$d$|S0|S1|Mean|
> |-|-|-|-|
> |18|77.84|74.24|76.04|
> |32|79.09|75.08|77.09|
> |**72**|**80.76**|**76.51**|**78.64**|
> |144|79.34|75.43|77.39|
>
> $d=72$ achieves the best trade-off between capacity and efficiency. MoPE temperature $\tau_mix$ (S3DIS 2-way 1-shot):
>
> |$\tau_{mix}$|S0|S1|Mean|
> |-|-|-|-|
> |0.2|78.46|75.11|76.78|
> |0.4|78.93|75.26|77.09|
> |0.6|79.71|75.67|77.69|
> |0.8|80.12|76.02|78.07|
> |**1.0**|**80.76**|**76.51**|**78.64**|
>
> The monotonically increasing trend confirms that a softer weighting distribution benefits expert utilization. Loss weight ablations are in Figure 4. We will add all above results to the final version.
>
> >> **Q5**: Reproducibility concern regarding RTX 5090 / PyTorch 2.8 / CUDA 13.0.
>
> **R5**: We have **successfully reproduced all results on RTX 4090** (PyTorch 1.13.1, CUDA 11.7, Python 3.9.0) — a widely accessible environment. The RTX 5090 is our primary development environment but is not required for reproduction. Regarding `pointnet2_ops_lib`, established compilation solutions are available on GitHub. We will update Appendix B accordingly and will open-source the code upon acceptance.
>
> >> **Q6**: The performance improvement of DPR-Net compared to DyPolySeg should be compared.
>
> **R6**: We thank the reviewer for this constructive suggestion and we will demonstrate the performance improvements compared to DyPolyNet in the revised version.

---

### Decision · Program_Chairs · 2026-04-30

**Decision:**

Accept (regular)

**Comment:**

This paper receives two weak accept recommendations and one reject recommendations. The two reviewers' with positive recommendations have acknowledged the response and achknowledge that the main concerns have been addressed. After reading the comments of the reviewer with negative recommendation and the response, the AC thinks that most of the concerns have been addressed.